# VanillaNet: the Power of Minimalism in Deep Learning

**Hanting Chen**[1], **Yunhe Wang**[1]*, **Jianyuan Guo**[1], **Dacheng Tao**[2]

[1] Huawei Noah's Ark Lab. [2] School of Computer Science, University of Sydney.

`{chenhanting,yunhe.wang,jianyuan.guo}@huawei.com, dacheng.tao@sydney.edu.au`

## Abstract

At the heart of foundation models is the philosophy of "more is different", exemplified by the astonishing success in computer vision and natural language processing. However, the challenges of optimization and inherent complexity of transformer models call for a paradigm shift towards simplicity. In this study, we introduce VanillaNet, a neural network architecture that embraces elegance in design. By avoiding high depth, shortcuts, and intricate operations like self-attention, VanillaNet is refreshingly concise yet remarkably powerful. Each layer is carefully crafted to be compact and straightforward, with nonlinear activation functions pruned after training to restore the original architecture. VanillaNet overcomes the challenges of inherent complexity, making it ideal for resource-constrained environments. Its easy-to-understand and highly simplified architecture opens new possibilities for efficient deployment. Extensive experimentation demonstrates that VanillaNet delivers performance on par with renowned deep neural networks and vision transformers, showcasing the power of minimalism in deep learning. This visionary journey of VanillaNet has significant potential to redefine the landscape and challenge the status quo of foundation model, setting a new path for elegant and effective model design. Pre-trained models and codes are available at `https://github.com/huawei-noah/VanillaNet` and `https://gitee.com/mindspore/models/tree/master/research/cv/vanillanet`.

## 1 Introduction

Over the past few decades, artificial neural networks have made remarkable progress, driven by the idea that increasing network complexity leads to improved performance. These networks, which consist of numerous layers with a large number of neurons or transformer blocks [50, 36], are capable of performing a variety of human-like tasks, such as face recognition [30], speech recognition [9], object detection [43, 17, 18], natural language processing [50], and content generation [3]. The impressive computational power of modern hardware allows neural networks to complete these tasks with both high accuracy and efficiency. As a result, AI-embedded devices are becoming increasingly prevalent in our lives, including smartphones, AI cameras, voice assistants, and autonomous vehicles.

Admittedly, one notable breakthrough in this field is the development of AlexNet [29], which consists of 12 layers and achieves state-of-the-art performance on the large-scale image recognition benchmark [8]. Building on this success, ResNet [23] introduces identity mappings through shortcut connections, enabling the training of deep neural networks with high performance across a wide range of computer vision applications, such as image classification [45], object detection [43], and semantic segmentation [38]. The incorporation of human-designed modules in these models, as well as the continued increase in network complexity, has undeniably enhanced the representational

---
*corresponding author

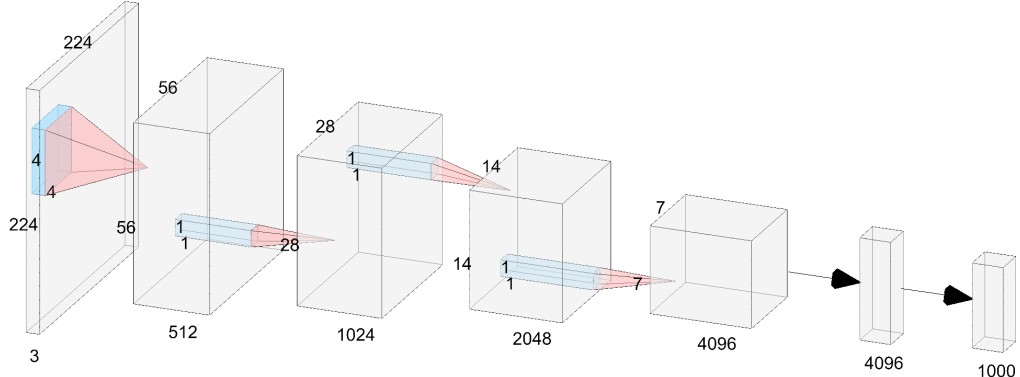

Figure 1: The architecture of VanillaNet-6 model, which consists of only 6 convolutional layers, which are very easily to be employed on any modern hardwares. The size of input features are downsampled while the channels are doubled in each stage, which borrows from the design of classical neural networks such as AlexNet [29] and VGGNet [45].

capabilities of deep neural networks, leading to a surge of research on how to train networks with more complex architectures [28, 24, 54] to achieve even higher performance.

Apart from convolutional architectures, Dosovitskiy et al.[14] have introduced the transformer architecture to image recognition tasks, demonstrating its potential for leveraging large-scale training data. Zhai et al.[60] investigated the scaling laws of vision transformer architectures, achieving an impressive 90.45% top-1 accuracy on the ImageNet dataset, which indicates that deeper transformer architectures, like convolutional networks, tend to exhibit better performance. Wang et al.[51] further proposed scaling the depth of transformers to 1,000 layers for even higher accuracy. Liu et al.[37] revisited the design space of neural networks and introduced ConvNext, achieving similar performance to state-of-the-art transformer architectures.

Although well-optimized deep and complex neural networks achieve satisfying performance, their increasing complexity poses challenges for deployment. For example, shortcut operations in ResNets consume significant off-chip memory traffic as they merge features from different layers [32]. Furthermore, complicated operations such as axial shift in AS-MLP [33] and shift window self-attention in Swin Transformer [36] require sophisticated engineering implementation, including rewriting CUDA codes.

These challenges call for a paradigm shift towards simplicity in neural network design. However, the development of ResNet has seemingly led to the abandonment of neural architectures with pure convolutional layers (without extra modules such as shortcuts). This is mainly due to the performance enhancement achieved by adding convolutional layers not meeting expectations. As discussed in [23], plain networks without shortcuts suffer from gradient vanishing, causing a 34-layer plain network to perform worse than an 18-layer one. Moreover, the performance of simpler networks like AlexNet [29] and VGGNet [45] has been largely outpaced by deep and complex networks, such as ResNets [23] and ViT [8]. Consequently, less attention has been paid to the design and optimization of neural networks with simple architectures. Addressing this issue and developing concise models with high performance would be of great value.

To this end, we propose VanillaNet, a novel neural network architecture emphasizing the elegance and simplicity of design while retaining remarkable performance in computer vision tasks. VanillaNet achieves this by eschewing excessive depth, shortcuts, and intricate operations such as self-attention, leading to a series of streamlined networks that address the issue of inherent complexity and are well-suited for resource-limited environments. To train our proposed VanillaNets, we conduct a comprehensive analysis of the challenges associated with their simplified architectures and devise a "deep training" strategy. This approach starts with several layers containing non-linear activation functions. As the training proceeds, we progressively eliminate these non-linear layers, allowing for easy merging while preserving inference speed. To augment the networks' non-linearity, we put forward an efficient, series-based activation function incorporating multiple learnable affine transfor-

mations. Applying these techniques has been demonstrated to significantly boost the performance of less complex neural networks. As illustrated in Figure 2, VanillaNet surpasses contemporary networks with elaborate architectures concerning both efficiency and accuracy, highlighting the potential of a minimalist approach in deep learning. This pioneering investigation of VanillaNet paves the way for a new direction in neural network design, challenging the established norms of foundation models and establishing a new trajectory for refined and effective model creation.

## 2  A Vanilla Neural Architecture

Over the past decades, researchers have reach some consensus in the basic design of neural networks. Most of the state-of-the-art image classification network architectures should consist of three parts: a stem block to transform the input images from 3 channels into multiple channels with downsampling, a main body to learn useful information, a fully connect layer for classification outputs. The main body usually have four stages, where each stage is derived by stacking same blocks. After each stage, the channels of features will expand while the height and width will decrease. Different networks utilize and stack different kinds of blocks to construct deep models.

Despite the success of existing deep networks, they utilize large number of complex layers to extract high-level features for the following tasks. For example, the well-known ResNet [23] requires 34 or 50 layers with shortcuts for achieving over 70% top-1 accuracy on ImageNet. The base version of ViT [14] consists of 62 layers since the query, key and value in self-attention require multiple layers to calculate.

With the growing of AI chips, the bottleneck of inference speed of neural networks would not be FLOPs or parameters, since modern GPUs can easily do parallel calculation with strong computing power. In contrast, their complex designs and large depths block their speed. To this end, we propose the vanilla network, *i.e.,* VanillaNet, whose architecture is shown in Figure 1. We follow the popular design of neural network with the stem, main body and fully connect layer. Different with existing deep networks, we only employ one layer in each stage to establish a extremely simple network with as few layer as possible.

Here we show the architecture of the VanillaNet in details, which takes 6 layers as an example. For the stem, we utilize a $4 \times 4 \times 3 \times C$ convolutional layer with stride 4 following the popular settings in [23, 36, 37] to map the images with 3 channels to features with $C$ channels. At stage 1, 2 and 3, a maxpooling layer with stride 2 is used to decrease the size and feature map and the number of channels is increased by 2. At stage 4, we do not increasing the number of channels as it follows an average pooling layer. The last layer is a fully connected layer to output the classification results. The kernel size of each convolutional layer is $1 \times 1$, since we aim to use minimal calculation cost for each layer while keep the information of feature maps. The activation function is applied after each $1 \times 1$ convolutional layer. To ease the training procedure of the network, batch normalization is also added after each layer. For the VanillaNet with different number of layers, we add blocks in each stage, which will be detailed in the supplementary material. It should be noted that the VanillaNet has no shortcut, since we empirically find adding shortcut shows little performance improvement. This also gives another benefit that the proposed architecture is extremely easy to implemented since there are no branch and extra blocks such as squeeze and excitation block [27].

Although the architecture of VanillaNet is simple and relatively shallow, its weak non-linearity caused limit the performance, Therefore, we propose a series of techniques to solve the problem.

## 3  Training of Vanilla Networks

It is common in deep learning to enhance the performance of models by introducing stronger capacity in the training phase [5, 57]. To this end, we propose to utilize a deep training technique to bring up the ability during training in the proposed VanillaNet, since deep network have stronger non-linearity than shallow network.

### 3.1  Deep Training Strategy

The main idea of deep training strategy is to train two convolutional layers with an activation function instead of a single convolution layer in the beginning of training procedure. The activation function is

gradually reduce to an identity mapping with the increasing number of training epochs. At the end of training, two convolutions can be easily merged into the one convolution to reduce the inference time. This kind of idea is also widely used in CNNs [11, 13, 10, 12]. We then describe how to conduct this technique in detail.

For an activation function $A(x)$ (which can be the usual functions such ReLU and Tanh), we combine it with an identity mapping, which can be formulated as:

$$A'(x) = (1 - \lambda)A(x) + \lambda x, \tag{1}$$

where $\lambda$ is a hyper-parameter to balance the non-linearity of the modified activation function $A'(x)$. Denote the current epoch and the number of deep training epochs as $e$ and $E$, respectively. We set $\lambda = \frac{e}{E}$. Therefore, at the beginning of training ($e = 0$), $A'(x) = A(x)$, which means the network have strong non-linearity. When the training converged, we have $A'(x) = x$, which means the two convolutional layers have no activation functions in the middle. We further demonstrate how to merge these two convolutional layers.

We first convert every batch normalization layer and its preceding convolution into a single convolution, which is a commonly used technique. We denote $W \in \mathbb{R}^{C_{out} \times (C_{in} \times k \times k)}, B \in \mathbb{R}^{C_{out}}$ as the weight and bias matrices of convolutional kernel with $C_{in}$ input channels, $C_{out}$ output channels and kernel size $k$. The scale, shift, mean and variance in batch normalization are represented as $\gamma, \beta, \mu, \sigma \in \mathbb{R}^{C_{out}}$, respectively. The merged weight and bias matrices are:

$$W_i' = \frac{\gamma_i}{\sigma_i} W_i, B_i' \quad = \frac{(B_i - \mu_i)\gamma_i}{\sigma_i} + \beta_i, \tag{2}$$

where subscript $i \in \{1, 2, ..., C_{out}\}$ denotes the value in $i$-th output channels.

After merging the convolution with batch normalization, we begin to merge the two $1 \times 1$ convolutions. Denote $x \in \mathbb{R}^{C_{in} \times H \times W}$ and $y \in \mathbb{R}^{C_{out} \times H' \times W'}$ as the input and output features, the convolution can be formulated as:

$$y = W * x = W \cdot \text{im2col}(x) = W \cdot X, \tag{3}$$

where $*$ denotes the convolution operation, $\cdot$ denotes the matrix multiplication and $X \in \mathbb{R}^{(C_{in} \times 1 \times 1) \times (H' \times W')}$ is derived from the im2col operation to transform the input into a matrix corresponding to the kernel shape. Fortunately, for $1 \times 1$ convolution, we find that the im2col operation becomes a simple reshape operation since there are no need for sliding kernels with overlap. Therefore, denote the weight matrix of two convolution layers as $W^1$ and $W^2$, the two convolution without activation function is formulated as:

$$y = W^1 * (W^2 * x) = W^1 \cdot W^2 \cdot \text{im2col}(x) = (W^1 \cdot W^2) * X, \tag{4}$$

Therefore, $1 \times 1$ convolution can merged without increasing the inference speed. It is worth noticed that although the deep training technique will increase the training FLOPs, it would not affect the inference cost, which is much more important when implementation.

## 3.2 Series Informed Activation Function

There have been proposed several different activation functions for deep neural networks, including the most popular Rectified Linear Unit (ReLU) and its variants (PReLU [22], GeLU [25] and Swish [42]). They focus on bring up the performance of deep and complex networks using different activation functions. However, as theoretically proved by the existing works [40, 15, 47], the limited power of simple and shallow network are mainly caused by the poor non-linearity, which is different with deep and complex networks and thus has not been fully investigated.

In fact, there are two ways to improve the non-linearity of a neural network: stacking the non-linear activation layers or increase the non-linearity of each activation layer, while the trend of existing networks choose the former one, which results in high latency when the parallel computation ability is excess.

One straight forward idea to improve non-linearity of activation layer is stacking. The serially stacking of activation function is the key idea of deep networks. In contrast, we turn to concurrently stacking the activation function. Denote a single activation function for input $x$ in neural network as

$A(x)$, which can be the usual functions such ReLU and Tanh. The concurrently stacking of $A(x)$ can be formulated as:

$$A_s(x) = \sum_{i=1}^{n} a_i A(x + b_i),\tag{5}$$

where $n$ denotes the number of stacked activation function and $a_i, b_i$ are the scale and bias (which are learned parameters) of each activation to avoid simple accumulation. The non-linearity of the activation function can be largely enhanced by concurrently stacking. Equation 5 can be regarded as a series in mathematics, which is the operation of adding many quantities.

To further enrich the approximation ability of the series, we enable the series based function to learn the global information by varying the inputs from their neighbors, which is similar with BNET [56]. Specifically, given a input feature $x \in \mathbb{R}^{H \times W \times C}$, where $H$, $W$ and $C$ are the number of its width, height and channel, the activation function is formulated as:

$$A_s(x_{h,w,c}) = \sum_{i,j \in \{-n,n\}} a_{i,j,c} A(x_{i+h,j+w,c}) + b_c,\tag{6}$$

where $h \in \{1, 2, ..., H\}$, $w \in \{1, 2, ..., W\}$ and $c \in \{1, 2, ..., C\}$. It is easy to see that when $n = 0$, the series based activation function $A_s(x)$ degenerates to the plain activation function $A(x)$, which means that the proposed method can be regarded as a general extension of existing activation functions. We use ReLU as the basic activation function to construct our series since it is efficient for inference in GPUs.

We further analyze the computation complexity of the proposed activation function compared with its corresponding convolutional layer. For a convolutional layer with $K$ kernel size, $C_{in}$ input channels and $C_{out}$ output channels, the computational complexity is:

$$\mathcal{O}(\text{CONV}) = H \times W \times C_{in} \times C_{out} \times k^2,\tag{7}$$

while computation cost of its series activation layer is:

$$\mathcal{O}(\text{SA}) = H \times W \times C_{out} \times (2n + 1)^2.\tag{8}$$

Therefore, we have:

$$\frac{\mathcal{O}(\text{CONV})}{\mathcal{O}(\text{SA})} = \frac{H \times W \times C_{in} \times C_{out} \times K^2}{H \times W \times C_{out} \times (2n + 1)^2} = \frac{C_{in} \times k^2}{(2n + 1)^2}.\tag{9}$$

Taking the 4th stage in VanillaNet-B as an example, where $C_{in} = 2048$, $k = 1$ and $n = 3$, the ratio is about 84. In conclusion, the computation cost of the proposed activation function is still much lower than the convolutional layers. More experimental complexity analysis will be shown in the following section.

It is noted that the proposed SIAF is significantly different with BNET [56]. The motivation of BNET is to improve BN's flexibility of fitting complex data distributions, which is resolved by introducing a linear transformation module with considering each neuron's neighborhood. In contrast, the motivation of the proposed SIAF is to enhance the non-linearity of shallow networks, which is implemented by combining multiple activation functions with different weights and biases. Empirically, as shown in Table 3 in our paper, the SIAF only works for shallow networks yet with huge improvements (>4.0% on ImageNet), while BNET is a general method with relatively little improvements ( 0.5% on ImageNet).

## 4 Experiments

In this section, we conduct experiments to verify the performance of the proposed VanillaNet on large scale image classification. Ablation study is provided to investigate effectiveness of each component of the proposed VanillaNet. We also visualize the feature of VanillaNet to further study how the proposed network learns from images.

### 4.1 Comparison with SOTA architectures

To illustrate the effectiveness of the proposed method, we conduct experiments on the ImageNet [8] dataset, which consists of $224 \times 224$ pixel RGB color images. The ImageNet dataset contains 1.28M

Table 1: Comparison on ImageNet. The depth denotes the number of non-linear layers. The Reall Acc denotes the accuracy with the ImageNet Real label [2]. Latency is tested on Nvidia A100 GPU with batch size of 1. When comparing with other baselines, we also apply the batch norm folding for fair comparison.

| Model | Params (M) | FLOPs (B) | Depth | Latency (ms) | Acc (%) | Real Acc (%) |
|---|---|---|---|---|---|---|
| MobileNetV3-Small [26] | 2.5 | 0.06 | 48 | 6.65 | 67.67 | 74.33 |
| MobileNetV3-Large [26] | 5.5 | 0.22 | 48 | 7.83 | 74.04 | 80.01 |
| ShuffleNetV2x1.5 [44] | 3.5 | 0.30 | 51 | 7.23 | 73.00 | 80.19 |
| ShuffleNetV2x2 [26] | 7.4 | 0.58 | 51 | 7.84 | 76.23 | 82.72 |
| RepVGG-A0 [13] | 8.1 | 1.36 | 23 | 3.22 | 72.41 | 79.33 |
| RepVGG-A1 [13] | 12.8 | 2.37 | 23 | 3.24 | 74.46 | 81.02 |
| RepVGG-B0 [13] | 14.3 | 3.1 | 29 | 3.88 | 75.14 | 81.74 |
| RepVGG-B3 [13] | 110.9 | 26.2 | 29 | 4.21 | 80.50 | 86.44 |
| ViTAE-T [55] | 4.8 | 1.5 | 67 | 13.37 | 75.3 | 82.9 |
| ViTAE-S [55] | 23.6 | 5.6 | 116 | 22.13 | 82.0 | 87.0 |
| ViTAEV2-S [62] | 19.2 | 5.7 | 130 | 24.53 | 82.6 | 87.6 |
| ConvNextV2-A [53] | 3.7 | 0.55 | 41 | 6.07 | 76.2 | 82.79 |
| ConvNextV2-F [53] | 5.2 | 0.78 | 41 | 6.17 | 78.0 | 84.08 |
| ConvNextV2-P [53] | 9.1 | 1.37 | 41 | 6.29 | 79.7 | 85.60 |
| ConvNextV2-N [53] | 15.6 | 2.45 | 47 | 6.85 | 81.2 | - |
| ConvNextV2-T [53] | 28.6 | 4.47 | 59 | 8.40 | 82.5 | - |
| ConvNextV2-B [53] | 88.7 | 15.4 | 113 | 15.41 | 84.3 | - |
| Swin-T [36] | 28.3 | 4.5 | 48 | 10.51 | 81.18 | 86.64 |
| Swin-S [36] | 49.6 | 8.7 | 96 | 20.25 | 83.21 | 87.60 |
| CMT-T [19] | 9.7 | 1.3 | 90 | 13.71 | 79.0 | 85.4 |
| CMT-S [19] | 26.26 | 4.05 | 135 | 23.09 | 83.4 | 88.2 |
| ResNet-18-TNR [52] | 11.7 | 1.8 | 18 | 3.12 | 70.6 | 79.4 |
| ResNet-34-TNR [52] | 21.8 | 3.7 | 34 | 5.57 | 75.5 | 83.4 |
| ResNet-50-TNR [52] | 25.6 | 4.1 | 50 | 7.64 | 79.8 | 85.7 |
| VanillaNet-5 | 15.5 | 5.2 | 5 | 1.61 | 72.49 | 79.66 |
| VanillaNet-6 | 32.5 | 6.0 | 6 | 2.01 | 76.36 | 82.86 |
| VanillaNet-7 | 32.8 | 6.9 | 7 | 2.27 | 77.98 | 84.16 |
| VanillaNet-8 | 37.1 | 7.7 | 8 | 2.56 | 79.13 | 85.14 |
| VanillaNet-9 | 41.4 | 8.6 | 9 | 2.91 | 79.87 | 85.66 |
| VanillaNet-10 | 45.7 | 9.4 | 10 | 3.24 | 80.57 | 86.25 |
| VanillaNet-11 | 50.0 | 10.3 | 11 | 3.59 | 81.08 | 86.54 |
| VanillaNet-12 | 54.3 | 11.1 | 12 | 3.82 | 81.55 | 86.81 |
| VanillaNet-13 | 58.6 | 11.9 | 13 | 4.26 | 82.05 | 87.15 |
| VanillaNet-13-1.5× | 127.8 | 26.5 | 13 | 7.83 | 82.53 | 87.48 |
| VanillaNet-13-1.5×[†] | 127.8 | 48.9 | 13 | 9.72 | 83.11 | 87.85 |

training images and 50K validation images with 1000 categories. We utilize strong regularization since the proposed VanillaNet has large number of parameters in each layer to capture useful information from the images with limited non-linearity. We also report the ImageNet Real results where the labels are refined. The latency is tested on Nvidia A100 GPU.

We propose architecture for VanillaNet with different number of layers. Table 1 shows the classification results on the ImageNet dataset using different networks. We list the number of parameters, FLOPs, depth, GPU latency and accuracy for comparison. In the past decades, researchers focus on minimize the FLOPs or the latency in ARM/CPU for portable networks since they assume that the computing power in edge devices is limited. As the development of modern AI chips, several mobile devices such as driverless vehicle [31] and robots [16] are required and able to carry multiple GPUs with huge computing power for seeking real-time feedback of external inputs. Therefore, we test the GPU latency with batch size 1, which means that the AI chip has enough computing power to calculate each network. Under this situation, we find that the inference speed has little relationship with the number of FLOPs and parameters. Taking MobileNetV3-Large a an example,

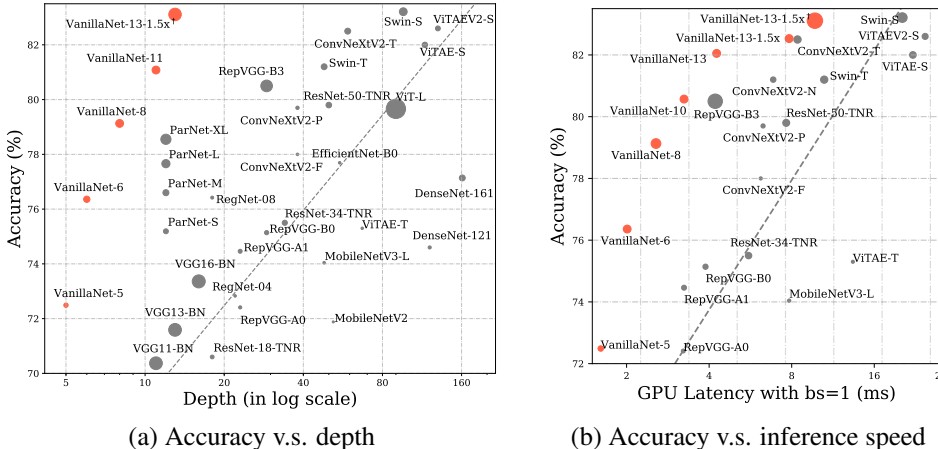

(a) Accuracy v.s. depth  (b) Accuracy v.s. inference speed

Figure 2: Top-1 Accuracy on ImageNet v.s. inference speed on Nvidia A100 GPU with batch size 1. Size of the circle is related to the depth and parameters of each architecture in (a) and (b), respectively. VanillaNet achieves comparable performance with deep neural networks while with much smaller depth and latency.

though it has a very low FLOPs (0.22B), its GPU latency is 7.83, which is even larger than our VanillaNet-13 with a 11.9B FLOPs. In fact, the inference speed in this setting is highly related to the complexity and number of layers. We can compare the inference speed of ShuffleNetV2x1.5 and ShuffleNetV2x2. In fact, their difference only lies in the number of channels. Therefore, although their number of parameters and FLOPs differs a lot. (0.3B v.s. 0.6B), their inference speed is nearly the same (7.23 and 7.84). We can also find in Table 1 that the straightforward architecture including ResNet, VGGNet and our VanillaNet without extra branch and complex blocks (*e.g.,* squeeze and excitation block or densely connects) achieves the highest inference speed.

To this end, we propose the VanillaNet, which is simple and has few convolutional layers without any branch (even without shortcut). We set different number of layers in VanillaNets to construct a series of networks. As shown in Table 1, the VanillaNet-9 achieves a 79.87% accuracy with only a 2.91ms inference speed in GPU, which is over 50% faster than the ResNet-50 and ConvNextV2-P with similar performance. The surprising result demonstrate the potential of VanillaNet in real-time processing over the existing deep networks. We also scale the number of channels and the pooling size to obtain the proposed VanillaNet-13-1.5×[†], which achieves an 83.11% Top-1 accuracy on ImageNet, which suggests that the proposed vanilla neural network still have power to obtain such a high performance on large scale image classification task. It is suggested that we may not need deep and complex networks on image classification since scaling up VanillaNets can achieve similar performance with deep networks. Although the proposed VanillaNet requires much more FLOPs and parameters, its inference speed in GPUs is much faster. In fact, the inference speed in GPUs and the number of parameters or FLOPs are two different aspects to measure the performance of neural networks. According to the "no free lunch" rule, it is nearly impossible to be satisfactory to both sides. Therefore, the proposed VanillaNet aims to the inference speed in GPUs and achieve much more higher speed then existing SOTA architectures.

The Figure 2 shows the depth and inference speed of different architectures. The inference speed with batch size 1 is highly related to the depth of the network instead of the number of parameters, which suggest that simple and shallow networks have huge potential in real-time processing. It can be easily find that the proposed VanillaNet achieve the best speed-accuracy trade-off among all these architectures with low GPU latency, which demonstrates the superiority of the proposed VanillaNet when the computing power is sufficient.

## 4.2 Ablation Study

In this section, we conduct ablation study to investigate the effectiveness of proposed modules, including the series activation function and the deep training technique. Besides, we analyze the influence of adding shortcuts in the proposed VanillaNet.

**Influence of number of series in activation function.** In the above section, we propose a series activation function to enhance the performance of plain activation function and enable global information exchange in feature maps. Table 2 shows the performance of the proposed VanillaNet using different number of $n$ in Equation 6. When $n = 0$, the activation function degenerate into the plain ReLU activation function. Although the inference speed of this network is

Table 2: Ablation study on the number of series.

| $n$ | FLOPs (B) | Latency (ms) | Top-1 (%) |
|---|---|---|---|
| 0 | 5.83 | 1.96 | 60.53 |
| 1 | 5.86 | 1.97 | 74.53 |
| 2 | 5.91 | 1.99 | 75.62 |
| 3 | 5.99 | 2.01 | 76.36 |
| 4 | 6.10 | 2.18 | 76.43 |

higher than using the series activation function, the network can only achieve a 60.53% top-1 accuracy on the ImageNet dataset, which cannot be applied in real-world applications. It proves that the poor non-linearity of activation function results in poor performance of vanilla networks.

To this end, we propose the series activation function. When $n = 1$, the network can achieve a 74.53% accuracy, which is a huge improvement compared with 60.53%. The result demonstrate the effectiveness of the proposed activation function. When the number of $n$ increases, the performance of the network brings up. We find that $n = 3$ is a good balance in the top-1 accuracy and latency. Therefore, we use $n = 3$ for the rest experiments. It should be noted that the FLOPs of the proposed activation function is very small compared with the original network, which is the same as the conclusion we derive in Equation 9.

**Influence of deep training.** As the VanillaNet is very shallow, we propose to increase the training non-linearity to bring up its performance. We then analyze the effectiveness of the proposed deep training technique. Table 3 shows the results on using deep training technique with VanillaNet-6. As a result, the original VanillaNet achieves a 75.23% top-1 accuracy, which is the baseline. By using the deep training technique, the proposed VanillaNet can achieve a 76.36% accuracy. The results demonstrate that the proposed deep training technique is useful for the shallow network.

Table 3: Ablation study on different networks.

| Network | Deep train. | Series act. | Top-1 (%) |
|---|---|---|---|
| VanillaNet-6 |  |  | 59.58 |
|  | ✓ |  | 60.53 |
|  |  | ✓ | 75.23 |
|  | ✓ | ✓ | 76.36 |
| AlexNet |  |  | 57.52 |
|  | ✓ |  | 59.09 |
|  |  | ✓ | 61.12 |
|  | ✓ | ✓ | 63.59 |
| ResNet-50 |  |  | 76.13 |
|  | ✓ |  | 76.16 |
|  |  | ✓ | 76.30 |
|  | ✓ | ✓ | 76.27 |

Moreover, we further apply the deep training and series activation function in other networks to show the generalization ability of the two techniques. Table 3 reports the results of AlexNet and ResNet-50, which are two classical deep neural networks, on the ImageNet dataset. The original AlexNet can only acheive a 57.52% accuracy with 12 layers. By applying the proposed deep training and series activation function, the performance of AlexNet can be largely brought up by about 6%, which demonstrates that the proposed technique is highly effective for shallow networks. When it turns to ResNet-50 whose architecture are relatively complex, the performance gain is little. This results suggests the deep and complex networks already have enough non-linearity without the proposed techniques.

**Influence of shortcuts.** In deep neural networks, a common sense is that adding shortcut can largely ease the training procedure and improve the performance [23]. To this end, we investigate whether shortcut would benefit the performance of shallow and simple network. We propose to use two kinds of location of shortcut, *i.e.,* shortcut after the activation function and shortcut before the activation function, which are proposed in the original ResNet [23] and PreAct-ResNet [24], respectively. Since the number

Table 4: Ablation on adding shortcuts.

| Type | Top-1 (%) |
|---|---|
| no shortcut | **76.36** |
| shortcut before act | 75.92 |
| shortcut after act | 75.72 |

of channels is large and the original convolution is with kernel size $1 \times 1$ in VanillaNet, adding a shortcut (even with $1 \times 1$ kernel size) would largely increase the FLOPs. Therefore, we use the

parameter-free shortcut. It should be noted that if the stride is 2, the parameter-free shortcut will use an average pooling to decrease the size of feature maps and if the number of channel is increasing, the parameter-free shortcut utilizes padding for the extra channels following the original setting in [23].

Table 4 shows the ablation study on adding shortcuts. We surprisingly find that using shortcuts, in spite of any type of shortcuts, has little improvement on the performance of the proposed VanillaNet. We suggest that the bottleneck of shallow networks is not the identity mapping, but the weak non-linearity. As reported in ResNet [23], the error rate of 18 layers' plain network (without shortcut) is 27.94%, which is similar to 18 layers' ResNet (27.88%). This result also suggests that shortcuts are not necessary for shallow networks. In fact, the shortcut is useless for bringing up the non-linearity and may decrease non-linearity since the shortcut skips the activation function to decrease the depth of the vanilla network, therefore results in lower performance.

**Performance of transfer learning.** To further demonstrate the generalization ability of the proposed VanillaNet, we fine-tune the proposed VanillaNet (pre-trained on ImageNet) on the well-known long-tailed classification dataset: iNaturalist 2018 [49] and follow the setting in Deit [48]. As shown in the Table 5, the performance of VanillaNet in pre-training scenarios is similar with other methods, which demonstrates its universality.

Table 5: Accuracy (%) on ImageNet and iNaturalist 2018.

| Model | ImageNet | iNat18 |
|---|---|---|
| ResNet-50 | 79.6 | 69.8 |
| VanillaNet-9 | 72.5 | 72.5 |
| Deit-B | 81.8 | 73.2 |
| VanillaNet-12 | 81.6 | 73.6 |

### 4.3 Experiments on COCO

Table 6: Performance on COCO detection and segmentation. FLOPs are calculated with image size (1280, 800)on Nvidia A100 GPU.

| Framework | Backbone | FLOPs | Params | FPS | $AP^b$ | $AP^b_{50}$ | $AP^b_{75}$ | $AP^m$ | $AP^m_{50}$ | $AP^b_{75}$ |
|---|---|---|---|---|---|---|---|---|---|---|
| RetinaNet [34] | Swin-T [36] | 245G | 38.5M | 27.5 | 41.5 | 62.1 | 44.2 | - | - | - |
| | VanillaNet-13-D | 397G | 75.4M | 29.8 | 43.0 | 62.8 | 44.3 | - | - | - |
| Mask RCNN [21] | Swin-T [36] | 267G | 47.8M | 28.2 | 42.7 | 65.2 | 46.8 | 39.3 | 62.2 | 42.2 |
| | ConvNextV2-N [53] | 221G | 35.2M | 34.4 | 42.9 | 65.5 | 46.9 | 39.6 | 62.5 | 42.2 |
| | VanillaNet-13-D | 421G | 77.1M | 32.6 | 42.9 | 65.5 | 46.9 | 39.6 | 62.5 | 42.2 |

To further demonstrate the effectiveness of the proposed VanillaNet on downstream tasks, we conduct evaluation in the COCO dataset [35]. We use RetinaNet [34] and Mask-RCNN [21] as the framework to evaluate the proposed method. FPS is measured on Nvidia A100 GPU. Details can be found in supplementary materials.

Table 6 shows the performance of the proposed VanillaNet on COCO detection and segmentation. The proposed VanillaNet can successfully achieve similar performance with the ConvNext and the Swin backbone. Although the FLOPs and Parameters of VanillaNet is much higher than Swin and ConvNext, it has much higher FPS, which demonstrates the effectiveness of vanilla architectures on object detection and instance segmentation tasks.

## 5 Conclusion

This paper fully investigates the feasibility of establishing neural networks with high performance but without complex architectures such as shortcut, high depth and attention layers, which embodies the paradigm shift towards simplicity and elegance in design. We present a deep training strategy and the series activation function for VanillaNets to enhance its non-linearity during both the training and testing procedures and bring up its performance. Experimental results on large scale image classification datasets reveal that VanillaNet performs on par with well-known deep neural networks and vision transformers, thus highlighting the potential of minimalism in deep learning. We will further explore better parameter allocation for efficient VanillaNet architectures with high performance. In summary, we prove that it is possible to achieve comparable performance with the state-of-the-art deep networks and vision transformers using a very concise architecture, which will unlock the potential of vanilla convolutiaonal network in the future.

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

## A  Network Architectures

The detailed architecture for VanillaNet with 7-13 layers can be found in Table 7, where each convolutional layer is followed with an activation function. For the VanillaNet-13-1.5×, the number of channels are multiplied with 1.5. For the VanillaNet-13-1.5×$^{\dagger}$, we further use adaptive pooling for stage 2,3 and 4 with feature shape 40×40, 20×20 and 10×10, respectively.

|  | Input | VanillaNet-5 | VanillaNet-6 | VanillaNet-7/8/9/10/11/12/13 |
|---|---|---|---|---|
| stem | 224×224 | | 4×4, 512, stride 4 | |
| stage1 | 56×56 | [1×1, 1024]×1
MaxPool 2×2 | [1×1, 1024]×1
MaxPool 2×2 | [1×1, 1024]×2
MaxPool 2×2 |
| stage2 | 28×28 | [1×1, 2048]×1
MaxPool 2×2 | [1×1, 2048]×1
MaxPool 2×2 | [1×1, 2048]×1
MaxPool 2×2 |
| stage3 | 14×14 | [1×1, 4096]×1
MaxPool 2×2 | [1×1, 4096]×1
MaxPool 2×2 | [1×1, 4096]×1/2/3/4/5/6/7
MaxPool 2×2 |
| stage4 | 7×7 | - | [1×1, 4096]×1 | [1×1, 4096]×1 |
| classifier | 7× 7 | | AvgPool 7×7
1×1, 1000 | |

Table 7: Detailed architecture specifications.

## B  Training Details

For classification on ImageNet, we train the VanillaNets for 300 epochs utilizing the cosine learning rate decay [39]. The $\lambda$ in Equ. 1 is linearly decayed from 1 to 0 on epoch 0 and 100, respectively. The training details can be fould in Table 8. For the VanillaNet-11, since the training difficulty is relative large, we use the pre-trained weight from the VanillaNet-10 as its initialization. The same technique is adopt for VanillaNet-12/13.

For detection and segmentation on COCO, we train all the networks using 12 epochs, multi-scale training augmentation and a linear learning rate decay for fair comparison. Following ConvNextV2 [53] which utilize self-supervised training, we use the ImageNet pre-trained weight using knowledge distillation [20] with $n = 4$ for a higher receptive field. We train the VanillaNet-13 using the Adamw optimizer with a batch size of 32, an initial learning rate of 8e-5 for RetinaNet and 1.3e-4 for Mask RCNN, an 0.05 weight decay and an 0.6 layer wise decay.

| Training Config | VanillaNet-{5/6/7/8/9/10/11/12/13} |
|---|---|
| weight init | trunc. normal (0.2) |
| optimizer | LAMB [58] |
| loss function | BCE loss |
| base learning rate | 3.5e-3 {5,8-13} /4.8e-3 {6-7} |
| weight decay | 0.35/0.35/0.35/0.3/0.3/0.25/0.3/0.3/0.3 |
| optimizer momentum | $\beta_1, \beta_2 = 0.9, 0.999$ |
| batch size | 1024 |
| training epochs | 300 |
| learning rate schedule | cosine decay |
| warmup epochs | 5 |
| warmup schedule | linear |
| dropout | 0.05 |
| layer-wise lr decay [6, 1] | 0 {5,8-12} /0.8 {6-7,13} |
| randaugment [7] | (7, 0.5) |
| mixup [61] | 0.1/0.15/0.4/0.4/0.4/0.4/0.8/0.8/0.8 |
| cutmix [59] | 1.0 |
| color jitter | 0.4 |
| label smoothing [46] | 0.1 |
| exp. mov. avg. (EMA) [41] | 0.999996 {5-10} /0.99992 {11-13} |
| test crop ratio | 0.875 {5-11} /0.95 {12-13} |

Table 8: ImageNet-1K training settings.

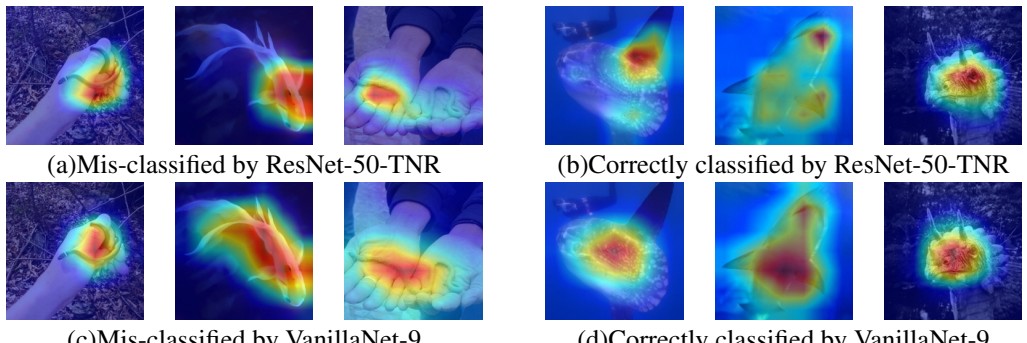

(a)Mis-classified by ResNet-50-TNR    (b)Correctly classified by ResNet-50-TNR

(c)Mis-classified by VanillaNet-9    (d)Correctly classified by VanillaNet-9

Figure 3: Visualization of attention maps of the classified samples by ResNet-50 and VanillaNet-9. We show the attention maps of their mis-classified samples and correctly classified samples for comparison.

## C   Visualization of Attention

To have a better understanding of the proposed VanillaNet, we further visualize the features using GradCam++ [4], which utilizes a weighted combination of the positive partial derivatives of the feature maps generated by the last convolutional layer with respect to the specific class to generate a good visual explanation.

Figure 3 shows the visualization results for VanillaNet-9 and ResNets-50-TNR [52] with similar performance. The red color denotes that there are high activation in this region while the blue color denotes the weak activation for the predicted class. We can find that these two networks have different attention maps for different samples. It can be easily found that for ResNet-50, the area of active region is smaller. For the VanillaNet with only 9 depth, the active region is much larger than that of deep networks. We suggest that VanillaNet may be strong in extract all relative activations in the input images and thoroughly extract their information by using large number of parameters and FLOPs. In contrast, VanillaNet may be weak on analyzing part of the useful region since the non-linearity is relatively low.

