# OpenReview forum: "VanillaNet: the Power of Minimalism in Deep Learning"
_NeurIPS.cc/2023/Conference — NeurIPS 2023 poster_

### Official Review · Reviewer_R2Cr · 2023-07-05

**Soundness:** 3 good
**Presentation:** 2 fair
**Contribution:** 3 good
**Rating:** 5
**Confidence:** 4

**Summary:**

This paper aims to minimize the complex design of modern deep models. The authors propose an architecture, VanillaNet, which eliminates high depth, residual connections, non-linear activations, self-attentions, and normalization layers, but during training, the model still use some components, such as activations and normalizations, to help model training. They also propose an activation function series to boost the capacity and performance of the models. Due to the simplicity, the model is ideal for resource-constrained environments and has lower latency than complex deep models. Their experiments show that the proposed model has similar performance than modern deep models.

**Strengths:**

1. The proposed model does not have high depth, residual connection, and normalization layers, which is beneficial for latency.
2. Despite the lack of complex design, the performance of the proposed idea is comparable to modern deep models.
3. The proposed activation function series offers performance boost with slightly more FLOPs and parameters.

**Weaknesses:**

1. It would be better to have a figure indicating where the merging operation describe in section 3.1 occurs in the model, so it is difficult to see where the merging opportunities are.
2. The idea of merging batch normalization with convolution describe between line 132 and line 137 is an already proposed inference acceleration called Batch Normalization Folding. Please make a clear distinction between what design are novel in this paper and what design are coming from existing work.
3. The content between lines 177-183 calculates the complexity (which directly tie to FLOPs) of the activation function series and claims the cost is very small, but in lines 90-91, the authors claims FLOPs is not important, so they are contradicting. Rather, it would make more sense to compare the latency of proposed activation and convolution.
4. The latency (profiled using a batch of size 1) in Table 4 could be misleading. In many inference cases, multiple instances can be batched together to improve throughput and make better usage of available hardwares, so it would be much better if throughput or latency (measured using reasonably large batch size) is also provided.
5. Code is not provided.

**Questions:**

The main concerns are listed above. In addition, is the proposed activation function series used in inference? If so, then the model still has activations and batch normalization (although the later one is folded into convolution), and compared ResNet, only the residual connections are removed. If not, does this mean all convolutions in each stage can be merged into a single convolutions?

This is an important question since the price of VanillaNet is that the parameter count and FLOPs need to be significantly higher to achieve similar performance compared to baselines.

**Limitations:**

There is no discussion on limitations. You should include a small limitation section. For example, you could include a discussion about the need for higher parameter count and FLOPs.

---

> ### Author Rebuttal · Authors · 2023-08-09
>
> **Q1:** Merging operation.
>
> **A1:** Sorry for the unclear statement. The inference architecture of VanillaNet can be found in the supplementary material. During the training time, each convolution layer is doubled into two convolutions with an activation between them. After training, the two convolutions will be merged into one convolution layer. We will provide a figure to explain this operation in detail in the final version.
>
> **Q2:** Clear distinction of designs.
>
> **A2:** Sorry for the misleading. The Batch Normalization Folding is coming from existing work and the others (deep training techniques and series informed activation functions) are designed in this paper. We will claim it in the final version.
>
> **Q3:** Compare the latency of proposed activation and convolution.
>
> **A3:** Thanks for the nice suggestion. As shown in the Table below, the latency of VanillaNet-6 using the proposed activation is 2.01, which is similar with that without the activation (1.96). Therefore, the latency cost of the activation is also very small, which is the same as the analysis by using FLOPs.
>
> | VanillaNet-6 | Acc(%) | Latency(ms) |
> |:---:|:---:|:---:|
> | with proposed activation | 76.36 | 2.01 |
> | without proposed activation | 60.53 | 1.96 |
>
>
> **Q4:** Provide latency measured using reasonably large batch size.
>
> **A4:** Thanks for the nice concern. We test the speeds using a 256 batchsize. The results show that the proposed VanillaNets can still achieve higher speed with similar performance in this setting.
>
> | name | Acc(%) | Latency(ms) on A100 with bs 256 |
> |:---:|:---:|:---:|
> | Swin-T | 81.18 | 96.69 |
> | ResNet-18 |70.6 | 31.24 |
> | ResNet-34 |75.5|54.06|
> | ResNet-50 |79.8|74.83|
> | ConvNextV2-P | 79.7|53.92|
> | ConvNextV2-N | 81.2| 70.41|
> | **VanillaNet-5** | 72.49 |22.64 |
> | **VanillaNet-6** | 76.36 |25.24|
> | **VanillaNet-9** | 79.87 |32.75|
> | **VanillaNet-12** | 81.55 |40.35|
>
> **Q5:** Code is not provided.
>
> **A5:** Thanks for the nice suggestion. We will provide code link in the final version.
>
> **Q6:** Is the proposed activation function series used in inference?
>
> **A6:** Yes. The model still has activations and batch normalization and all convolutions in each stage cannot be merged into a single convolutions. Compared with ResNet, we introduce the series informed activation function and the deep training technique. As a result, the proposed VanillaNet achieves much higher inference speed compared with ResNet.

---

> > ### Author Response · Authors · 2023-08-15
> >
> > Dear reviewer R2Cr:
> >
> > We sincerely thank you for the review and comments.
> >
> > We have provided corresponding responses and results, which we've tried our best to cover your concerns. With extra experiments during the rebuttal period, the advantage of our approach becomes clearer. Some missing comparison have been provided and will be added to the final paper.
> >
> > Please let us know whether your concerns have been well addressed. We would like to further discuss with you if you still have any unclear parts of our work.
> >
> > Best,
> >
> > The Authors

---

> > > ### Comment · Reviewer_R2Cr · 2023-08-19
> > >
> > > Thank you for the efforts on the response. The response addressed most of my concerns.
> > >
> > > Additional comments:
> > > I am assuming that the batch norm folding is not applied to other baselines when doing efficiency comparison (if it is used in baselines, it would be helpful to make it clear in the updated draft).
> > > Since the batch norm folding technique is a common inference acceleration and can also be applied to other baselines that contains batch norms, please include an efficiency comparison to baselines with batch norm folding.
> > >
> > > Assuming the response and above comparison will be incorporated into the final version. I will raise my score.

---

> > > > ### Author Response · Authors · 2023-08-19
> > > >
> > > > Sorry for the misleading. When comparing with other baselines, we also apply the batch norm folding for fair comparison. We will make it clear in the final verison. Thanks for the nice suggestion.

---

### Official Review · Reviewer_jvnA · 2023-07-06

**Soundness:** 3 good
**Presentation:** 3 good
**Contribution:** 3 good
**Rating:** 6
**Confidence:** 2

**Summary:**

The paper presents VanillaNet, a neural network architecture that emphasizes simplicity by avoiding high depth, shortcuts, and complex operations like self-attention. The proposed VanillaNet performs comparably to well-known deep neural networks and vision transformers.

**Strengths:**

**Originality**: To the best of my knowledge, the proposed VanillaNet is novel.

**Quality**:     This paper is of high quality since it proposed a new model architecture and conducted extensive experiments on ImageNet and ablation studies to demonstrate its effectiveness.

**Clarity**: The paper is well-structured and clear, providing detailed explanations of the VanillaNet architecture and the proposed techniques.

**Significance**: The paper is significant as it paves the way for a new direction in neural network design.

**Weaknesses:**

-  Although the proposed VanillaNet reduced the latency a lot, the required Params and FLOPs increased a lot. e.g. FOR SWIN-S and VannillaNet that have comparable accuracy on ImageNet, the proposed vanillaNet needs 127.8 M and 48.9 FLOPs while Swin-S only needs 49.6 M and 8.7B FLOPs.

**Questions:**

- Since pre-training significantly impacts the model's performance, it would be better to see how the proposed model architecture performs in pre-training scenarios.

**Limitations:**

the authors adequately addressed the limitations.

---

> ### Author Rebuttal · Authors · 2023-08-09
>
> **Q1:** More parameters and FLOPs.
>
> **A1:** Thanks for the nice concern. The inference speed in GPUs and the number of parameters or FLOPs are two different aspects to measure the performance of neural networks. According to the “no free lunch” rule, it is nearly impossible to be satisfactory to both sides. Therefore, the proposed VanillaNet aims to the inference speed in GPUs and achieve much more higher speed then existing SOTA architectures.
>
> **Q2:** Performance in pre-training scenarios.
>
> **A2:** We fine-tune the proposed VanillaNet (pre-trained on ImageNet) on the well-known long-tailed classification dataset: iNaturalist 2018 [1] and follow the setting in Deit [2].
>
> | Model | ImageNet Acc. | iNat18 Acc. |
> |:---:|:---:|:---:|
> | ResNet-50 | 79.6 | 69.8 |
> | VanillaNet-9 | 72.49 | 72.45 |
> | Deit-B | 81.8 | 73.2 |
> | VanillaNet-12 | 81.6 | 73.6 |
>
> As shown in the Table above, the performance of VanillaNet in pre-training scenarios is similar with other methods, which demonstrates its universality. We will include these results in the final version.
>
> *[1] The inaturalist challenge 2018 dataset. Arxiv 2018.*
>
> *[2] Training data-efficient image transformers & distillation through attention. PMLR 2021.*

---

### Official Review · Reviewer_xYue · 2023-07-06

**Soundness:** 3 good
**Presentation:** 3 good
**Contribution:** 4 excellent
**Rating:** 7
**Confidence:** 5

**Summary:**

The paper introduces VanillaNet, a neural network architecture that emphasizes simplicity in design while maintaining remarkable performance in computer vision tasks. VanillaNet achieves this by avoiding excessive depth, shortcuts, and intricate operations like self-attention. The authors propose a "deep training" strategy to train VanillaNets, which starts with several layers containing non-linear activation functions and progressively eliminates them as training proceeds. Additionally, an efficient, series-based activation function incorporating multiple learnable affine transformations is introduced to augment the networks' non-linearity. The experimental results show that VanillaNet achieves comparable or even better performance than state-of-the-art deep neural networks and vision transformers on various computer vision tasks while being more efficient and resource-friendly.


**Strengths:**

1. This paper is well-written and organized, with clear explanations of the proposed VanillaNet architecture. The paper also provides detailed experimental setups and results, making it easy to understand and reproduce the experiments.
2. The proposed VanillaNet is novel. By avoiding excessive depth, shortcuts, and intricate operations like self-attention, VanillaNet offers a streamlined and efficient design. This simplicity not only makes VanillaNet well-suited for resource-limited environments but also challenges the prevailing notion that increasing network complexity leads to improved performance, which may have implications for future research in the field.
3. The experimental results are convincing. Despite its simplicity and low depth, VanillaNet delivers performance on par with renowned deep neural networks and vision transformers. For example, a 13 layers' VanillaNet achieves an 83% top-1 accuracy on ImageNet.


**Weaknesses:**

1. While VanillaNet's simplicity and low depth make it an attractive option for efficient deployment in GPUs, one weakness of the proposed method is that it has significantly more parameters and FLOPs than some existing methods. It is noted that VanillaNet has a large number of parameters in each layer to capture useful information from the images with limited non-linearity. This can make it more challenging to train and deploy VanillaNet on certain devices with limited computational resources.
2. The proposed deep training strategy is used to enhance the performance of VanillaNet by introducing stronger capacity during training, which suggests that the proposed method requires more training FLOPs.


**Questions:**

1. Table 3 shows that shortcuts are useless for the proposed VanillaNets. As the shortcuts are one of the most popular components in modern networks, it would be better to provide more explainations about this phenomenon.
2. As shown in the suppmentary material, the VanillaNet-13 is achieved by adding convolutions on the 3rd stage of VanillaNet-7, what is the designed logic behind?


**Limitations:**

yes.

---

> ### Author Rebuttal · Authors · 2023-08-09
>
> **Q1:** More parameters and FLOPS.
>
> **A1:** Thanks for the nice concern. The inference speed in GPUs and the number of parameters or FLOPs are two different aspects to measure the performance of neural networks. According to the “no free lunch” rule, it is nearly impossible to be satisfactory to both sides. Therefore, the proposed VanillaNet aims to the inference speed in GPUs and achieve much more higher speed then existing SOTA architectures.
>
> **Q2:** More training FLOPs.
>
> **A2:** Thanks for the suggestion. Although the deep training technique will increase the training FLOPs, it would not affect the inference cost, which is much more important when implementation.
>
> **Q3:** Explanation on shortcuts.
>
> **A3:** We suggest that the bottleneck of vanilla networks is not the identity mapping, but the weak non-linearity. The shortcut is useless for bringing up the non-linearity. As reported in ResNet [1], the error rate of 18 layers’ plain network is 27.94%, which is similar to 18 layers’ ResNet (27.88%). This result also suggests that shortcuts are not necessary for shallow networks.
>
> *[1] Deep Residual Learning for Image Recognition. CVPR 2016.*
>
> **Q4:** Designed logic.
>
> **A4:** Thanks for the nice concern. We follow the designed logic in ResNet [1] to assign extra layers in the 3rd stages for simplify, which can be further optimized in the future work.

---

### Official Review · Reviewer_prVa · 2023-07-07

**Soundness:** 4 excellent
**Presentation:** 3 good
**Contribution:** 3 good
**Rating:** 5
**Confidence:** 4

**Summary:**

This paper introduces VanillaNet, a neural network architecture that emphasizes simplicity in design. By discarding commonly used ingredients, e.g., high depth, shortcuts, and self-attention mechanism, VanillaNet is refreshingly concise. VanillaNet series have only 5-13 layers. The authors devise the “deep training” strategy, in which nonlinear activation functions are pruned after training to restore the original architecture. They also leverage series-based activation function to incorporate multiple learnable affine transformations. The experiments on classification and detection demonstrate the effectiveness of the proposed VanillaNet.

**Strengths:**

Motivation: This paper is different from the previous research direction of the backbone architecture, abandoning the exploration of the residual connection or deepen the network. It shows the potential of the shallow network. The proposed VanillaNet avoids complex operations and make the structure more compact and easy to understand.

Quality: The paper is clear and easy to follow.

Originality: The “deep training technique” proposed in this paper is well motivated and effective. The activation function is gradually reduced to an identity mapping during training. And finally two convolutions can be easily merged into one convolution. I like this idea. In addition, the “series activation function” which enables global information exchange also seems effective.

Experiment: The authors conduct sufficient experiments and ablation studies to demonstrate the effectiveness of the proposed VanillaNet. VanillaNet can achieve comparable performance to current SOTA models on several computer vision tasks with faster inference speed

Overall, the motivation challenges the “more is different” foundational model philosophy, opening up new possibilities for efficient deployment of deep learning models.

**Weaknesses:**

About the application of “deep training technique”. It seems that this technique can only be apllied to BN. If the normalization layer is LN, it cannot be merged.


About the number of parameters. Although the inference speed of VanillaNet is superior to other methods, the heavy number of paramters and “deep train” strategy may increase the training cost of the network. According to my understanding, the amount of network parameters in training stage should be much larger than the paratmers shown in Table 4 (in inferece stage).

**Questions:**

Inference speed: The authors report the latency on GPU and pytorch. However, in more scenarios (or platform), such as tensorRT, will the gap between VanillaNet and other networks become smaller?

Series Informed Activation Function: What’s the difference between SIAF and BNET? I think these two methods should be explained in more details.

**Limitations:**

Yes

---

> ### Author Rebuttal · Authors · 2023-08-09
>
> **Q1:** The number of parameters.
>
> **A1:** Thanks for the suggestion. Although the deep training technique will increase the training cost, it would not affect the inference cost, which is much more important when implementation.
>
> **Q2:** Inference speed:
>
> **A2:** Thanks for the nice concern. We test the speeds using tensorRT. The results show that the proposed VanillaNets achieve higher speed with similar performance.
>
> | name | Acc(%) | Latency(ms) using TRT on A100 |
> |:---:|:---:|:---:|
> | Swin-T | 81.18 | 1.41 |
> | ResNet-18 |70.6 | 0.41 |
> | ResNet-34 |75.5|0.77|
> | ResNet-50 |79.8|0.80|
> | ResNet-101 |81.3|1.58 |
> | ResNet-152 |81.8|2.30 |
> | **VanillaNet-5** | 72.49 |0.33 |
> | **VanillaNet-6** | 76.36 |0.40|
> | **VanillaNet-9** | 79.87 |0.58|
> | **VanillaNet-12** | 81.55 |0.75|
> | **VanillaNet-13** | 82.05|0.82|
>
> **Q3:** Difference between SIAF and BNET.
>
> **A3:** The motivation of BNET is to improve BN's flexibility of fitting complex data distributions, which is resolved by introducing a linear transformation module with considering each neuron's neighborhood. In contrast, the motivation of SIAF is to enhance the non-linearity of shallow networks, which is implemented by combining multiple activation functions with different weights and biases. Empirically, as shown in Table 2 in our paper, the SIAF only works for shallow networks yet with huge improvements (>4.0% on ImageNet), while BNET is a general method with relatively little improvements (~0.5% on ImageNet). Moreover, SIAF can combined with BNET to further improve the networks’ performance. We will include more analysis in the final version.

---

### Official Review · Reviewer_qK4N · 2023-07-15

**Soundness:** 3 good
**Presentation:** 3 good
**Contribution:** 4 excellent
**Rating:** 8
**Confidence:** 4

**Summary:**

The authors propose a minimalist neural network backbone architecture as an alternative to current ResNets and ViTs. The minimalism is focussed on faster inference, avoiding complex operations with higher time complexity when implemented on modern GPUs. In their proposed architecture, individual operations are limited to convolutions, activation functions and linear projections, avoiding more time-complex operations like skip connections or self-attention. The resulting 6-layer network surpass latest CNN and ViT backbones at a fraction of inference latency on ImageNet top-1 accuracy metrics. A novel deep training strategy (merge two consecutive layers post-training) and series activation functions (stacked activation functions for more complex non-linearity under lower time complexity) are introduced to facilitate better learning by these minimalist shallow networks. In addition to the base architecture, a series of scaled-up versions provide increasing performance, still at minimal inference latency.

**Strengths:**

1) Most of the paper is extremely well written with good logical flow, strong motivation for each proposal, and clear explanations of methodology.
2) Proposed deep training and series activation functions appear novel, well-motivated, and interesting.
3) Thorough ablations and discussion clearly demonstrate motivation for each design choice
4) Strong results over SOTA backbones for classification, detections, and segmentation
5) Detailed training details (in supp.) supports reproducibility of results

**Weaknesses:**

1) Missing details in series activation function: it appears scale and bias terms (Eq 5, 6) are learned parameters - please explicitly mention this or otherwise explain how these values are selected. Assuming former, how are these values initialized? Does the initialization affect final outcome (maybe conduct some ablation)? What are the final values for these parameters that are learned across runs (do they collapse in any occasion? do they convey any information about data distribution)? Can these parameters be fixed at start instead of learning?
2) Writing (minor): grammatical errors in numerous places across sections 3 & 4, captions of figures not descriptive enough or self-contained (e.g. Table 4 - explain terms like depth, real-acc vs acc in caption - reader needs to skim through main text to understand headers of table).
3) More evaluations could strengthen the paper (and convince about universality of new backbone): maybe include results on some long-tailed classification datasets, more than one detection and segmentation dataset.
4) Section 4.2 & Figure 2 (attention map) does not appear to convey anything useful in current form. Consider updating description / main text to discuss one of the examples in detail (or maybe move to supplementary).

**Questions:**

See 1 in weaknesses

**Limitations:**

No clear discussion of limitations.

---

> ### Author Rebuttal · Authors · 2023-08-09
>
> **Q1:** Missing details in series activation function.
>
> **A1:** Sorry for the unclear claim. The scale and bias terms are learned parameters. We initialize them using Kaiming initialization [1]. We also test other initialization. As shown in the Table below, the initialization affect little final outcome (<0.2%).
>
> | VanillaNet-5 | Kaiming init[1] | Xavier init[2] | Gaussian init |
> |:---:|:---:|:---:|:---:|
> | Accuracy | 72.49 | 72.45 | 72.36 |
>
> The final values differs across runs. The reason is that deep network is highly non-convex and the converged values rely on the initialized values (which is different across runs). Therefore, these parameters cannot be fixed at start.
>
> *[1] Delving Deep into Rectifiers: Surpassing Human-Level Performance on ImageNet Classification. ICCV 2015.*
>
> *[2] Understanding the difficulty of training deep feedforward neural networks. AISTATS 2010.*
>
> **Q2:** Writing.
>
> **A2:** Sorry for the mistake. We will fix the grammatical errors and enrich the caption of tables to make them self-contained.
>
> **Q3:** More evaluations.
>
> **A3:** Thanks for the suggestion. We conduct experiments on the well-known long-tailed classification dataset: iNaturalist 2018 [3] and follow the setting in Deit [4].
>
> | Model | ImageNet Acc. | iNat18 Acc. |
> |:---:|:---:|:---:|
> | ResNet-50 | 79.6 | 69.8 |
> | VanillaNet-9 | 72.49 | 72.45 |
> | Deit-B | 81.8 | 73.2 |
> | VanillaNet-12 | 81.6 | 73.6 |
>
> As shown in the Table above, the performance of VanillaNet in long-tailed dataset is similar with that in classical dataset compared with other methods, which demonstrates its universality. We will include these results in the final version.
>
> *[3] The inaturalist challenge 2018 dataset. Arxiv 2018.*
>
> *[4] Training data-efficient image transformers & distillation through attention. PMLR 2021.*
>
> **Q4:** Attention map.
>
> **A4:** Thanks for the suggestion. We will move them to supplementary and include more discussion in the final version.

---

> > ### Comment · Reviewer_qK4N · 2023-08-15
> > **Post Rebuttal**
> >
> > Thanks for the detailed rebuttal.
> >
> > Assuming updates to final manuscript in accordance to rebuttal, I raise my vote further. I strongly support acceptance for this paper.

---

### Decision · Program_Chairs · 2023-09-21

**Decision:**

Accept (poster)

**Comment:**

In this paper, the authors present a simple convolutional neural network architecture for computer vision problems that matches many modern architectures advancements in the last few years. The goal of this architecture is to aim for faster inference, and avoid complex operations such as skip connections, higher depths, and self-attention. The resulting architecture is demonstrated on image classification with ImageNet, and object detection with COCO achieving favorable performance in light of inference speed. The authors achieve this through a training strategy in which they slowly adapt a nonlinearity in between two layers towards an identity function. At the end of training, the two subsequent convolutions may be folded into a single linear operation for fast inference.

The reviewers commented positively on the simplicity of the method, the overall performance both in terms of latency and predictive performance, and the overall novelty. The reviewers also indicated concerns about the lack of methodological details, the increased number of model parameters relative to baseline models, and the limitation of the method to Batch Normalization (and not LayerNorm). Given that there remained no consensus in the reviewer scores, I considered this paper to be borderline acceptance. The paper shows strong performance and results but I am concerned that the scope of impact for these results may be limited due to the fact that these models must be CNN's with BatchNorm. The research community has generally moved away from this architecture style in order to provide additional functionality not available in standard CNN architectures. That said, given the simplicity and strength of these results and in spite of these limitations, I have decided to accept this paper to conference as I think it would be a great reminder that “older” architecture designs still have strong merit and should not be abandoned.